# GRADIENT DESCENT AND ATTENTION MODELS: CHALLENGES POSED BY THE SOFTMAX FUNCTION

## ABSTRACT

Transformers have become ubiquitous in modern machine learning applications, yet their training remains a challenging task often requiring extensive trial and error. Unlike previous architectures, transformers possess unique attention-based components, which can complicate the training process. The standard optimization algorithm, Gradient Descent, consistently underperforms in this context, underscoring the need for a deeper understanding of these difficulties: existing theoretical frameworks fall short and fail to explain this phenomenon. To address this gap, we analyze a simplified Softmax attention model that captures some of the core challenges associated with training transformers. Through a local analysis of the gradient dynamics, we highlight the role of the Softmax function on the local curvature of the loss and show how it can lead to ill-conditioning of these models, which in turn can severely hamper the convergence speed. Our experiments confirm these theoretical findings on the critical impact of Softmax on the dynamics of Gradient Descent.

## 1 INTRODUCTION

In recent years, transformer architectures have demonstrated remarkable success in various applications from natural language processing (Devlin et al., 2019; Achiam et al., 2023) to computer vision (Dosovitskiy et al., 2020). However, despite their impressive performance, our theoretical understanding of transformer training remains limited. For example, Stochastic Gradient Descent (SGD), which has been a staple optimization algorithm for deep learning models, fails to train transformers effectively (Liu et al., 2020). This has led to the belief that there are unique elements in the transformer architecture and the associated loss landscape that introduce challenges distinct from those in other architectures like Convolutional Neural Networks (CNNs).

Recently, several papers have attempted to explain this phenomenon through comparisons of SGD with adaptive methods. Indeed, empirical observations indicate that while SGD outperforms adaptive methods on tasks such ImageNet classification with CNNs (Wilson et al., 2017), the opposite is true for transformer training. Liu et al. (2020) claimed that this failure of SGD is due to vanishing and imbalanced gradients observed in experiments. Meanwhile, Jiang et al. (2023) introduced a variant of the condition number and empirically showed that it is larger on the path taken by SGD than the one taken by Adam for standard transformer architectures. Zhang et al. (2020) hypothesized that the performance gap might be due to the heavy-tailed and non-Gaussian nature of the stochastic gradient noise. However, Kunstner et al. (2023) challenged this explanation by showing that even in the full-batch case, SGD performs poorly compared to adaptive methods. Although these works highlight interesting features of transformer training that are different for Adam compared with SGD, their theoretical explanations of the source of these differences remain very limited.

In parallel, there has been a line of work that has focused on theoretically analyzing the dynamics of gradient descent on transformers and providing global convergence guarantees without explaining its poor performance. For instance, Tarzanagh et al. (2023b) showed that gradient descent on a simplified attention model solves a max-margin problem and provided guarantees for global and local convergence in the binary classification setting. Meanwhile, Abbe et al. (2023) showed that the incremental learning phenomenon, whereby singular values are learned one-by-one, is present in transformers. However, these works did not provide any rates of convergence that would shed light on the slow or lack of convergence of SGD in practice. Others provided rates of convergence but

were limited to heavily overparameterized settings (Wu et al., 2023) and assumed pretrained weights Liu et al. (2023).

Clearly, there exists a gap between the current theoretical analysis of Gradient Descent and the empirical observations demonstrating its difficulty in optimizing transformers. This leads to the following question:

*Why does Gradient Descent perform poorly on attention models?*

## 1.1 PAPER CONTRIBUTIONS

In this paper, we identify a critical aspect of attention models that contributes to the difficulties gradient descent faces in training them efficiently: the preconditioning effects induced by the Jacobian of the Softmax function. Our analysis focuses on a simplified one-layer Softmax attention model, previously introduced by Tarzanagh et al. (2023b); Oymak et al. (2023), which captures key challenges encountered in training more complex attention architectures. We provide sufficient conditions for GD to converge linearly in the overparameterized case, but show that this setting, which has also been the focus of prior works Wu et al. (2023), fails to explain the poor performance of gradient descent in practice. Motivated by this, we analyze the more realistic underparameterized case and highlight the preconditioning role of the Jacobian of the softmax. Our analysis of the local dynamics around stationary points reveals that the ill-conditioning of the Jacobian of the Softmax function, which occurs when the softmax probabilities are far from uniform, can significantly hinder the convergence of gradient descent. Specifically, we show that the Softmax parameterization alters the condition number of the Hessian of the loss by a multiplicative factor equal to the square of the ratio of the largest to smallest attention probabilities, thus significantly slowing convergence to sparse attention matrices.

## 2 RELATED WORK

**Dynamics of Attention.** Understanding the dynamics of attention has been the focus of many works in recent years. One line of work was interested in how transformers learn meaningful representations. For example, Li et al. (2023c) analyzed the optimization dynamics of a single-layer transformer and showed how it learns "semantic structure" from text data. Snell et al. (2021) showed that a single attention head learns to focus on salient words while Jelassi et al. (2022) focused on image data and showed how a simplified Vision transformer (ViT) can learn spatial structure without any inductive bias of spatial locality. Another line of work, that is closer to our work, focused on understanding the optimization dynamics of gradient descent. Tarzanagh et al. (2023a) and Tarzanagh et al. (2023b) analyzed a similar simplified attention model trained with a decreasing loss and established asymptotic convergence of gradient descent, showing that the solutions along some regularization paths solve an SVM problem. Abbe et al. (2023) considered a single-head transformer with diagonal attention weights and small initialization, and they showed that the model displays the incremental learning phenomenon in which the learned weights gradually increase in rank. These works did not provide any rates of convergence and instead focused on the implicit bias/regularization from the gradient descent algorithm. Liu et al. (2023) considered a shallow transformer trained with the hinge loss using an initialization from a pretrained model and analyzed its sample complexity. Although they consider an underparameterized setting in which the width of the model is of order at least $\log(N)$ where $N$ is the number of data samples, their analysis is restricted to a specific data model in which the tokens can be split into relevant and irrelevant sets. Moreover, their focus is on generalization properties and the only result that pertains to the optimization dynamics of Gradient Descent is limited to an analysis of the concentration of attention weights at a sublinear rate during training. Wu et al. (2023) is the most similar to some of our results, namely Theorem 1. They conduct a similar analysis on a one-layer self-attention model and provide sufficient conditions for linear convergence under $\mathcal{O}(N)$ overparameterization, but do not address the underparameterized setting. Moreover, their analysis relies on the final linear output layer to guarantee linear convergence, whereas our analysis yields a linear rate convergence based solely on the attention weights in the layer inside the Softmax function.

**Failure of SGD on transformers.** Several works have explored why SGD performs poorly on transformers. Zhang et al. (2020) hypothesized that it is caused by the heavy-tailed stochastic noise in language tasks. However, Kunstner et al. (2023) found that the SGD fails even in the full-batch case.

Moreover, Zhang et al. (2024) showed that Vision transformers trained with SGD on ImageNet suffer from the same problems. Therefore, the stochasticity of the gradients and the data modality cannot explain the failure of SGD on transformers. Pan & Li (2023) introduce the notion of directional sharpness and argue that SGD has a high directional sharpness, which they claim is correlated with low performance of optimization algorithms. Jiang et al. (2023) propose a new notion of condition number and show that it is large on the path taken by SGD. Zhang et al. (2024) explain the poor performance of SGD by computing the spectrum of the Hessian and showing that the spectrum across blocks varies significantly for transformers, unlike other architectures like CNNs and MLPs. Their theoretical results are however limited to quadratic models. Finally, Yadav et al. (2023) hypothesize that the poor performance of SGD is due to the heavy-tailed class imbalance. They empirically show that other architectures such as CNNs suffer from the same issue under heavy-tailed class imbalance.

## 3 Preliminaries

**Notation.** For any integer $n \geq 1$, we denote by $[n]$ the set $\{1, \ldots, n\}$. We use lower-case and upper-case bold letters to represent vectors and matrices, respectively. The $i$-th entry of a vector $\mathbf{x}$ is denoted as $\mathbf{x}_i$. For a vector valued function of time $\mathbf{u}(t)$, we let $\dot{\mathbf{u}} = \dot{\mathbf{u}}(t) = \frac{d}{dt}\mathbf{u}(t)$. We denote the Euclidean norm of a vector $\mathbf{x}$ by $\|\mathbf{x}\|_2$. We use $\sigma_{min}(\mathbf{A})$ (resp. $\sigma_{max}(\mathbf{A})$) and $\lambda_{min}(\mathbf{A})$ (resp. $\lambda_{max}(\mathbf{A})$) to denote the minimum (resp. maximum) singular value and the minimum (resp. maximum) eigenvalue of a matrix $\mathbf{A}$. We use $\Delta^n$ to denote the $n$-dimensional probability simplex in $\mathbb{R}^{n+1}$ and $\mathring{\Delta}^n$ to denote its interior. $T\Delta^n$ denotes the tangent space of the n-dimensional probability simplex. $\phi(\cdot)$ denotes the Softmax transformation. For any vector $\mathbf{x} \in \mathbb{R}^n$, we use $\mathrm{diag}(\mathbf{x})$ to denote the diagonal $n \times n$ matrix whose diagonal entries are $\mathbf{x}_1, \ldots, \mathbf{x}_n$. We use $\mathbb{1}_n$ to denote the vector of ones in $\mathbb{R}^n$. $\delta_{ij}$ denotes the Kronecker delta and is equal to 1 when $i = j$ and 0 otherwise.

**Attention.** A central component in the transformer architecture is the attention mechanism (Vaswani et al., 2017). It is designed to capture long-range interactions between three types of input vectors: queries, keys, and values, that can each be stacked together in matrices $\mathbf{Q} \in \mathbb{R}^{m \times d_{qk}}, \mathbf{K} \in \mathbb{R}^{n \times d_{qk}}$, and $\mathbf{V} \in \mathbb{R}^{n \times d_v}$ respectively. In this work, we focus on the dot-product attention defined as:

$$\mathrm{Attention}(\mathbf{Q}, \mathbf{K}, \mathbf{V}) = \phi(\mathbf{Q}\mathbf{K}^\top)\mathbf{V} \tag{1}$$

where the Softmax function $\phi : \mathbb{R}^n \to \Delta^{n-1}, \phi(\mathbf{w})_i = \frac{e^{\mathbf{w}_i}}{\sum_{j=1}^n e^{\mathbf{w}_j}}$ acts row-wise on its argument when it is a matrix.

In transformers, a standard attention layer takes as input two matrices $\mathbf{X} \in \mathbb{R}^{n \times d}$ and $\mathbf{Z} \in \mathbb{R}^{m \times d}$. The query, key, and value matrices are obtained via linear transformations of $\mathbf{X}$ and $\mathbf{Z}$

$$\mathbf{Q} = \mathbf{Z}\mathbf{W}_Q, \quad \mathbf{K} = \mathbf{X}\mathbf{W}_K, \quad \mathbf{V} = \mathbf{X}\mathbf{W}_V \tag{2}$$

where $\mathbf{W_K} \in \mathbb{R}^{d \times d_{qk}}, \mathbf{W_Q} \in \mathbb{R}^{d \times d_{qk}}$, and $\mathbf{W_V} \in \mathbb{R}^{d \times d_v}$ are learnable weight matrices.

**Self-Attention** is a particular case of attention in which the queries, keys and values are all obtained from the same input matrix, i.e. $\mathbf{X} = \mathbf{Z}$.

**Tunable tokens.** In practice, some of the query tokens are sometimes also learned. For example, a [CLS] or prompt token is typically appended to the input features of a model for the purpose of classification or to adapt the model to new tasks. The latter is referred to as prompt tuning and has been introduced as a more efficient alternative to fine-tuning the transformer weights (Lester et al., 2021; Liu et al., 2023). Furthermore, Oymak et al. (2023) identified scenarios in which prompt attention, which corresponds to freezing the selt-attention weights and tuning the prompt token, is more expressive than self-attention. Following (Oymak et al. (2023); Tarzanagh et al. (2023b)), we will therefore consider a simplified attention model with one tunable token $\mathbf{p} \in \mathbb{R}^d$ and a value vector $\mathbf{v} \in \mathbb{R}^d$. The key and query weights are combined in one matrix $\mathbf{W} = \mathbf{W}_Q \mathbf{W}_K^\top$. This model outputs a scalar which can be used for classification or regression:

$$f(\mathbf{X}) = \phi(\mathbf{X}\mathbf{W}\mathbf{p})^\top \mathbf{X}\mathbf{v} \in \mathbb{R} \tag{3}$$

**Problem Setting.** Given a training dataset $(\mathbf{X}_i, y_i)_{i=1}^N$ where $\mathbf{X}_i \in \mathbb{R}^{n \times d}$ and $y_i \in \mathbb{R}$ for all $i \in [N]$, we consider the empirical minimization problem

$$\mathcal{L}(\mathbf{W}, \mathbf{p}, \mathbf{v}) := \frac{1}{N} \sum_{i=1}^N \ell(f(\mathbf{X}_i), y_i), \tag{4}$$

where $\ell : \mathbb{R} \times \mathbb{R} \to \mathbb{R}$ is differentiable and convex in the first variable. The model is trained via Gradient Descent (GD). We make the following assumptions throughout the paper.

**Assumption 1.** *The weights $\mathbf{W}$ and $\mathbf{v}$ are fixed and only $\mathbf{p}$ is trained.*

**Rationale behind Assumption 1.** Following Tarzanagh et al. (2023b), we can simplify our analysis by freezing one of the parameters, either $\mathbf{p}$ or $\mathbf{W}$, which play symmetric roles in the dynamics within the softmax function. In this work, we choose to freeze $\mathbf{W}$ and train $\mathbf{p}$, aligning our approach with the framework of prompt tuning.

Furthermore, we assume a two-stage optimization process that aligns with empirical observations showing that the value weights $\mathbf{W_V}$ in transformers are learned significantly faster than the attention weights $\mathbf{W_Q}$ and $\mathbf{W_K}$ (Li et al., 2023b). One explanation for this phenomenon is that the gradients of $\mathbf{W_V}$ involving attention weights sum to one, resulting in larger updates, whereas the gradients for $\mathbf{W_Q}$ and $\mathbf{W_K}$ are proportional to these matrices themselves, which are near zero at initialization and hence update more slowly.

**Assumption 2.** *The input data satisfies $\mathbf{K}_i \mathbf{K}_j^\top = \delta_{ij} I_n$ for all $i, j \in [N]$.*

**Remark 1.** *Assumption 2 requires $d \geq n$. This is already true in many applications. For example, in language tasks, the number of tokens $n$ can be the length of a sentence, which is typically smaller than the dimensions of tokens (for example $d = 512$). A similar assumption on the orthogonality of the data has been made in prior work such as Wu et al. (2023).*

In what follows, we will thus write the loss as a function of $\mathbf{p}$ only:

$$\mathcal{L}(\mathbf{p}) = \frac{1}{N} \sum_{i=1}^{N} \ell(\phi(\mathbf{K}_i \mathbf{p})^\top \mathbf{X}_i \mathbf{v}, y_i), \ \ \mathbf{K}_i = \mathbf{X}_i \mathbf{W}. \tag{5}$$

**Jacobian of the Softmax function.** For any vector $\mathbf{z}$ in $\mathbb{R}^n$, let $\mathbf{J}(\mathbf{z})$ be the $n$ by $n$ matrix given by:

$$\mathbf{J}(\mathbf{z}) = \text{diag}(\mathbf{z}) - \mathbf{z}\mathbf{z}^\top = \text{diag}(\mathbf{z})(\mathbf{I}_n - \mathbb{1}\mathbf{z}^\top), \tag{6}$$

then the Jacobian of the Softmax function $\phi : \mathbb{R}^n \to \Delta^{n-1}$ can be written as:

$$\frac{d\phi(\mathbf{w})}{d\mathbf{w}} = \mathbf{J}(\phi(\mathbf{w})). \tag{7}$$

The matrix-valued function $\mathbf{J}$ will play an important role in our analysis of the gradient dynamics, we will therefore state some of its useful properties when restricted to $\mathring{\Delta}^{n-1}$:

**Lemma 1.** *Let $\mathbf{z}$ be a vector in $\mathring{\Delta}^{n-1}$. The matrix $\mathbf{J}(\mathbf{z})$ satisfies the following:*

1. *$\mathbf{J}(\mathbf{z})$ is a symmetric positive semidefinite matrix.*

2. *The vector $\mathbb{1}_n$ is the eigenvector associated with the eigenvalue $0$, i.e. $\mathbf{J}(\mathbf{z})\mathbb{1}_n = 0$.*

3. *Let $\lambda_1(\mathbf{z}) \leq \cdots \leq \lambda_n(\mathbf{z})$ denote the eigenvalues of $\mathbf{J}(\mathbf{z})$ and let $\tilde{\mathbf{z}}$ be a vector whose entries are the entries of $\mathbf{z}$ sorted in ascending order $\tilde{\mathbf{z}}_i \leq \tilde{\mathbf{z}}_{i+1}$ for $i \in [n-1]$. The eigenvalues satisfy*

$$0 = \lambda_1(\mathbf{z}) < \tilde{\mathbf{z}}_1 \leq \lambda_2(\mathbf{z}) \leq \cdots \leq \lambda_n(\mathbf{z}) \leq \tilde{\mathbf{z}}_n < 1. \tag{8}$$

## 4 MAIN RESULTS

In this section, we present our central findings on the dynamics of gradient descent when applied to training softmax attention models through two distinct yet complementary perspectives.

In the first subsection, we examine the behavior of gradient descent within an overparameterized setting ($d \geq N$). By leveraging classical Polyak-Łojasiewicz (PL) and smoothness assumptions, we establish a linear convergence rate. This analysis underscores the efficiency of optimization in scenarios where the model is overparameterized. Although it reveals a degradation in the convergence rate as attention vectors become sparser, the analysis fails to account for cases that are observed in practice and does not explicitly capture the mechanism underlying the suboptimal performance

of gradient descent. To address these limitations, the second subsection explores gradient descent dynamics without assuming overparameterization. Due to the inherent complexities and the current lack of robust technical tools for a global analysis in this regime, we focus on a local analysis around stationary points. This investigation reveals that the softmax mechanism can induce severe ill-conditioning, particularly when attention becomes sparse. Such ill-conditioning significantly impedes convergence, resulting in slower optimization progress.

Together, these analyses provide a nuanced understanding of gradient descent dynamics in softmax attention models, highlighting both the strengths of overparameterized settings and the challenges faced in more realistic, constrained parameter regimes.

## 4.1 GLOBAL CONVERGENCE IN THE OVERPARAMETERIZED CASE

Here, we establish the global convergence of gradient descent using a common strategy that relies on two main ingredients: the Lipschitz continuity of the gradient of the loss, and the Polyak-Lojasiewicz (PL) inequality (Polyak, 1963). These two conditions only need to hold along the trajectories, as referenced in Nguyen (2021), and Wu et al. (2023).

**Definition 1.** *Let $\mu > 0$. A differentiable function $f : \mathbb{R}^p \to \mathbb{R}^+$ is said to satisfy the $\mu$-Polyak-Łojasiewicz ($\mu$-PŁ) inequality over a subset $U \subset \mathbb{R}^p$ if for all $\mathbf{u} \in U$*

$$\frac{1}{2}\|\nabla f(\mathbf{u})\|^2 \geq \mu \left( f(\mathbf{u}) - \inf_{\mathbf{u}' \in U} f(\mathbf{u}') \right) \tag{9}$$

We consider the training of $\mathbf{p}$ via GD while $\mathbf{W}$ and $\mathbf{v}$ are frozen, which yields the following update equation

$$\mathbf{p}_{t+1} = \mathbf{p}_t - \frac{\eta}{N} \sum_{i=1}^{N} \mathbf{K}_i^\top \mathbf{J}(\phi(\mathbf{K}_i \mathbf{p}_t)) \mathbf{X}_i \mathbf{v}_t \nabla \ell(f(\mathbf{X}_i), y_i). \tag{10}$$

In this subsection, we assume that $\ell$ is $m$-strongly convex and $L$-smooth, and we introduce the following useful time-varying matrix

$$\mathbf{F}(\mathbf{p}_t) = \frac{1}{N} \begin{bmatrix} (\mathbf{K}_1^\top \mathbf{J}(\phi(\mathbf{K}_1 \mathbf{p}_t)) \mathbf{X}_1 \mathbf{v})^\top \\ \vdots \\ (\mathbf{K}_N^\top \mathbf{J}(\phi(\mathbf{K}_N \mathbf{p}_t)) \mathbf{X}_N \mathbf{v})^\top \end{bmatrix} \in \mathbb{R}^{N \times d} \tag{11}$$

We can thus rewrite the gradient update equation in the following simplified form

$$\mathbf{p}_{t+1} = \mathbf{p}_t + \eta \mathbf{F}(\mathbf{p}_t)^\top \mathbf{g}(\mathbf{p_t}), \tag{12}$$

where $\mathbf{g}(\mathbf{p_t})$ is the vector of gradients of $\ell$ with respect to the output of the model, i.e. its $i$-th component is $\mathbf{g_i}(\mathbf{p_t}) = \nabla \ell(f(\mathbf{X}_i), y_i)$. The following lemma, which will be used to provide a PŁ inequality in our convergence proof, now follows straightforwardly from the identity $\nabla \mathcal{L}(\mathbf{p}) = \mathbf{F}(\mathbf{p})^\top \mathbf{g}(\mathbf{p})$.

**Lemma 2.** *The loss function $\mathcal{L}$ satisfies a pointwise inequality:*

$$\frac{1}{2}\|\nabla \mathcal{L}(\mathbf{p})\|_2^2 \geq \mu(\mathbf{p})(\mathcal{L}(\mathbf{p}) - \mathcal{L}^*), \tag{13}$$

*where $\mu(\mathbf{p}) = 2m\sigma_{min}^2(\mathbf{F}(\mathbf{p}))$ for all $\mathbf{p} \in \mathbb{R}^d$.*

The next theorem provides sufficient conditions to guarantee the linear convergence of GD. The conditions establish a uniform lower bound for $\mu(\mathbf{p})$ which depends on the initialization of the weight vector $\mathbf{p}$. Additionally, they ensure that the solutions remain bounded, thereby guaranteeing the Lipschitz continuity of the gradients.

**Theorem 1.** *Suppose $d \geq N$, and let $c_1$ and $c_2$ be some positive constants such that $c_1 < 1$ and $c_2 < \min(2, 1/c_1)$. Define the following quantities:*

$$C_\sigma = \max_{1 \leq i \leq N} \sigma_{max}(\mathbf{X}_i), \quad L' = \sigma_{max}^2(W) C_\sigma^3 L \|\mathbf{v}\| \left( 2C_\sigma \|\mathbf{v}\| + 3\sqrt{\frac{2(\mathcal{L}_0 - \mathcal{L}^*)}{m}} \right)$$

$$\mu = \frac{c_1}{L'}, \quad \eta = \frac{c_2}{L'}$$

$$C_p = \sqrt{\frac{2(\mathcal{L}_0 - \mathcal{L}^*)}{mN}} \frac{\sigma_{max}(W) C_\sigma^2 \|\mathbf{v}\| c_2 L}{L'(1 - \sqrt{1 - c_1 c_2})}, \quad M_p = \|\mathbf{p}_0\| + C_p$$

$$\gamma = \frac{1}{\sqrt{N}} (\min(2, 3C_\sigma \sigma_{max}(\mathbf{W}) C_p) C_\sigma^2 \sigma_{max}(\mathbf{W}) \|\mathbf{v}\|_2)$$

*If $\sigma_{min}(\mathbf{F}(\mathbf{p}_0)) > \gamma + \sqrt{\frac{\mu}{2m}}$, then gradient descent on the function $\mathcal{L}$ with step size $\eta$ converges at a linear rate:*

$$\mathcal{L}(t) - \mathcal{L}^* \leq (1 - \eta\mu)^t (\mathcal{L}(0) - \mathcal{L}^*), \quad \text{for all } t \geq 0. \tag{14}$$

*Moreover, the weights $\mathbf{p}$ and $\mathbf{v}$ remain bounded throughout the trajectory*

$$\|\mathbf{p}_t\|_2 \leq M_p, \quad \text{for all } t \geq 0.$$

**Proof sketch:** The proof is by induction. At every iteration $t$, we show that the weights are bounded by the constants $M_p$. Next, we establish smoothness of the loss along the trajectories and show that the smallest singular value of $\mathbf{F}$ is bounded away from zero. This shows that the loss satisfies the PL inequality with constant $\mu$ at each iteration. The proof resembles other convergence proofs that establish a uniform lower bound on the PL constant and Lipschitz smoothness along trajectories Nguyen (2021); Wu et al. (2023). However, it does not rely on a final linear layer to prove that the PL condition is satisfied along the trajectories.

**Remark 2.** *Note that $\mathbf{F}(\mathbf{p}_0)$ is an $N \times d$ matrix. Therefore, the case $d < N$ would guarantee $\sigma_{min}(\mathbf{F}(\mathbf{p}_0)) = 0$ and the bound becomes vacuous. However, even in the case $d \geq N$, the theorem does not guarantee linear convergence and additional assumptions on the data, the initialization and the frozen weights are needed to satisfy the conditions stated in the theorem.*

**Remark 3.** *Identical proof techniques can be applied to establish linear convergence even when $\mathbf{v}$ is not held constant. However, to focus on how the dynamics of the softmax function influence the convergence of gradient descent, we present the case with $\mathbf{v}$ fixed. This simplification facilitates a clearer comparison with the results discussed in the subsequent subsection. Moreover, it shows that linear convergence can be obtained without relying on a trainable final linear layer such as in Wu et al. (2023).*

In order to guarantee linear convergence as stated in the previous theorem, we need the following assumptions:

**Assumption 3.** *The input data matrices $\mathbf{X}_i$ are surjective for all $i = 1, \dots N$.*

**Assumption 4.** *The matrix $\mathbf{W}$ has full rank.*

**Remark 4.** *In the case of multiple heads, the matrix $\mathbf{W}$ has low rank by construction. However, since we are considering a single head, it is reasonable to assume that $\mathbf{W}$ has full rank.*

**Proposition 1.** *Suppose that the data samples $X_i$ are drawn from a distribution $\mathcal{D}$. Let $\mathbf{v} = \alpha \mathbf{u}$ with $\|\mathbf{u}\|_2 = 1$. There exists $A > 0$ such that if $\alpha > A$ and assumptions 2, 3, 4 are satisfied, then $\mathbb{P}_{\mathcal{D}}(\sigma_{min}(\mathbf{F}(\mathbf{p}_0)) > \gamma + \sqrt{\mu/2m})) = 1$.*

**Limitations of the overparameterized setting.** One significant drawback of this style of analysis, which relies on the PL inequality, is that it requires that the norm of $\mathbf{p}$ remain bounded along its trajectory. However, to learn sparse or approximately sparse attention maps, the norm of the weight vector $\mathbf{p}$ must approach infinity. As the bound $C_p$ increases to allow for that, the convergence rate described in the theorem worsens and eventually yields a vacuous bound. Moreover, the ability to analyze gradient descent behavior in contexts where learned attention probabilities are approximately sparse is critical. The primary objective of the attention mechanism is to focus on the most relevant parts of the input, resulting in attention probabilities that are typically far from uniform. This has been confirmed by prior empirical observations (Wu et al., 2023; Li et al., 2023a; Chen et al., 2021), and can also be clearly observed across layers and heads in the attention weights that we included in Appendix E for the Vision Transformer model (Dosovitskiy et al., 2020) used in our experiments.

### 4.2 LINEAR STABILITY IN THE UNDERPARAMETERIZED CASE

In practice, the number of data points typically exceeds the dimensions of the tokens. For example, in the original transformer paper (Vaswani et al., 2017), the base model uses tokens of dimension $512$, but is trained on datasets containing millions of samples. More generally, all the layers in the architecture have widths that do not exceed 4096. As a result, it is crucial to explore the *underparameterized* setting where $d \ll N$.

The underparameterized setting has been virtually untouched in the deep learning optimization literature, where overparameterization is invariably an essential component of analysis in enabling proofs of a local PL inequality and hence linear convergence Allen-Zhu et al. (2019); Du et al. (2019); Nguyen (2021); Bombari et al. (2022). Without access to these technical tools, we will focus on a more tractable local analysis around stationary points.

We start by rewriting the gradient descent update equation:

$$\mathbf{p}_{t+1} = \mathbf{p}_t - \frac{\eta}{N} \sum_{j=1}^{N} \mathbf{K}_j^\top \mathbf{J}(\phi(\mathbf{K}_j \mathbf{p}_t)) \nabla \tilde{\ell}_j(\phi(\mathbf{K}_j \mathbf{p}_t)). \tag{15}$$

where we set $\tilde{\ell}_j(\mathbf{x}) := \ell(\mathbf{x}^\top \mathbf{X}_j \mathbf{v}, y_j)$ for ease of presentation.

And we can write the induced dynamics on the attention vectors for each $i = 1, \ldots, N$ :

$$\phi(\mathbf{K}_i \mathbf{p}_{t+1}) = \phi\Big( \mathbf{K}_i \mathbf{p}_t - \frac{\eta}{N} \sum_{j=1}^{N} \mathbf{K}_i \mathbf{K}_j^\top \mathbf{J}(\phi(\mathbf{K}_j \mathbf{p}_t)) \nabla \tilde{\ell}_j(\phi(\mathbf{K}_j \mathbf{p}_t)) \Big)$$

$$= \phi\Big( \mathbf{K}_i \mathbf{p}_t - \frac{\eta}{N} \mathbf{J}(\phi(\mathbf{K}_i \mathbf{p}_t)) \nabla \tilde{\ell}_i(\phi(\mathbf{K}_i \mathbf{p}_t)) \Big)$$

where the last equality follows from Assumption 2. Since the dynamics are identical for all $i = 1, \ldots N$, we drop the subscript $i$ and set $\mathbf{z}_t = \phi(\mathbf{K}_i \mathbf{p}_t)$. Next, using the Taylor formula with Lagrange remainder, we know that there exists $\mathbf{c}_t$ in the segment joining $\mathbf{z}_t$ to $\mathbf{z}_{t+1}$ such that:

$$\mathbf{z}_{t+1} = \mathbf{z}_t - \frac{\eta}{N} \mathbf{J}^2(\mathbf{z}_t) \nabla \tilde{\ell}(\mathbf{z}_t) + \frac{\eta^2}{2N^2} D^2 \phi(\mathbf{c}_t)(\mathbf{J}(\mathbf{z}_t) \nabla \tilde{\ell}(\mathbf{z}_t), \mathbf{J}(\mathbf{z}_t) \nabla \tilde{\ell}(\mathbf{z}_t)), \tag{16}$$

where the $i$-th coordinate of $D^2 \phi(\mathbf{x})(\mathbf{h}, \mathbf{h})$ is:

$$[D^2 \phi(\mathbf{x})(\mathbf{h}, \mathbf{h})]_i = \sum_{j,k} \frac{\partial^2 \phi_i(\mathbf{x})}{\partial x_j \partial x_k} \mathbf{h}_j \mathbf{h}_k. \tag{17}$$

The next theorem characterizes the stationary points of the dynamics in equation 16 that are in the interior of the simplex. In particular, we show that by linearizing the system around these stationary points, we can obtain a lower bound on the condition number for the local linear dynamics. This bound grows as we get close to boundaries of the simplex, which characterize the sparse attention solutions.

**Theorem 2.** *Let $\mathbf{z}^*$ be a stationary point of the dynamics described in equation 16. Suppose that $\mathbf{z}^*$ is in the interior of the simplex, then the following holds:*

1. *The linearization of the system around $\mathbf{z}^*$ is given by $\zeta_{t+1} = \zeta_t - \frac{\eta}{N} \mathbf{J}^2(\mathbf{z}^*) \nabla^2 \tilde{\ell}(\mathbf{z}^*) \zeta_t$.*

2. *Let $\mu = \lambda_{min}(\nabla^2 \tilde{\ell}(\mathbf{z}^*))$ and $L = \lambda_{max}(\nabla^2 \tilde{\ell}(\mathbf{z}^*))$. Suppose that $\mu > 0$ and let $\kappa$ be the condition number of $\mathbf{J}^2(\mathbf{z}^*) \nabla^2 \tilde{\ell}(\mathbf{z}^*)$ when restricted to the tangent space of the simplex $T\Delta^{n-1} = \{\mathbf{u} \in \mathbb{R}^n : \mathbb{1}^\top \mathbf{u} = 0\}$. Then we have the following lower bound on $\kappa$*

$$\kappa \geq \frac{\mu \max_i(\mathbf{z}_i^*)^2}{L \min_i(\mathbf{z}_i^*)^2}. \tag{18}$$

**Proof Sketch:** The first statement is obtained using standard linearization techniques which rely on only considering the linear part in the Taylor expansion formula applied to the system around $\mathbf{z}^*$. This yields an equation of the form $\zeta_{t+1} = \zeta_t - \frac{\eta}{N} D\mathbf{V}(\mathbf{z}^*) \zeta_t$ where $V(\mathbf{x}) = \mathbf{J}(\mathbf{x}) f(\mathbf{x}) = -\mathbf{J}^2(\mathbf{x}) \nabla \tilde{\ell}(\mathbf{x})$

and the dynamics are restricted to $T\Delta^{n-1}$. We first use the elementwise product rule to compute $D\mathbf{V}$ as a function of $Df$, then $Df$ as a function of $\nabla^2\tilde{\ell}$. We use the fact that $\mathbf{z}^*$ is an equilibrium to obtain $D\mathbf{V}(\mathbf{z}^*)P = -\mathbf{J}^2(\mathbf{z}^*)\nabla\tilde{\ell}(\mathbf{z}^*)P$, where $P$ is the orthogonal projection onto the $T\Delta^{n-1}$. The proof of the second statement requires establishing bounds on the largest and smallest eigenvalues of $DV(\mathbf{z}^*)$ restricted to the tangent space of the simplex using some basic linear algebra arguments. The full proof is provided in Appendix D.

**Remark 5.** *Note that convexity of the loss $\tilde{\ell}$ is not required for any of the results in Theorem 2 to hold. Moreover, the Hessians $\nabla^2\tilde{\ell}$ and $\nabla^2\ell$ are related via the choice of the vector $\mathbf{v}$ and the data point $\mathbf{X}$.*

**Remark 6.** *A main consequence of the theorem is that the condition number grows as $\mathbf{z}^*$ gets close to the boundary of the simplex. In fact, if we set $\epsilon = \min_i(\mathbf{z}_i^*)$, then we have that $\max_i(\mathbf{z}_i^*) \geq \frac{1-\epsilon}{n-1}$. As a result, the bound in the previous theorem implies that $\kappa \geq \frac{\mu(1-\epsilon)^2}{L(n-1)^2\epsilon^2}$ which goes to infinity as $\epsilon$ approaches 0. This shows that the local convergence rate of the linearized system that is derived in Theorem 2 gets worse as the attention vectors become sparser. Moreover, we see that this is induced by the growth of the condition number of the Softmax Jacobian.*

## 5 EXPERIMENTAL RESULTS

In this section, we validate our theoretical findings and interpretations regarding the role of the softmax and sparse attention through a series of experiments.

### 5.1 SIMPLIFIED ONE-LAYER SOFTMAX ATTENTION MODEL

First, we consider the training of the model defined in equation 3 using a squared loss which satisfies the conditions for linear stability in Theorem 2 as well as the strong convexity and Lipschitz smoothness used in Theorem 1. Both $\mathbf{p}$ and $\mathbf{v}$ are initialized with a Gaussian distribution $\mathcal{N}(0, 1/d)$, and our data matrices $\mathbf{X}_i$ are drawn from a standard Gaussian distribution with $n = d$. The labels $y_i$ are generated using identical teacher models whose weights are drawn from a Gaussian distribution with variances $\sigma_\mathbf{v}^2 = 1$ and $\sigma_\mathbf{p}^2 = \{0.1, 1\}$. We assume that $\mathbf{W} = I_d$ and train the models with full batch gradient descent on $\mathbf{p}$ and $\mathbf{v}$ on a dataset of $N = 100$ samples. Figure 1 shows the evolution of the training loss over 10 runs for four settings with different values of $d = \{20, 150, 400\}$. For each setting, we compute the average ratio across runs and data points of the largest to smallest attention probabilities at the end of training, which we denote by $R$. This ratio is equal to 1 when the probabilities are uniform and grows as the distribution becomes more peaked. Note that a large $R$ implies a larger condition number around stationary points in Theorem 2. The teacher models are designed to have different values of $R$ which we tune via the variance $\sigma_\mathbf{p}$. Namely, in the four curves shown in Figure 1, the blue curve in the left plot, which corresponds to the successful training of GD in the underparameterized setting, was produced by training on labels generated from a teacher model with $R_{teacher} \approx 10$ corresponding to $\sigma_\mathbf{p} = 0.1$. The other three curves were all trained using teacher models with $R_{teacher} \approx 10^{25}$ corresponding to $\sigma_\mathbf{p} = 1$.

Figure 1 shows that in the underparameterized setting, GD performs poorly when $R$ is very large, i.e. when the stationary points are arbitrarily close to boundaries of the simplex, as predicted by our theory. However, when $R$ is small, gradient descent successfully optimizes the training loss. In the overparameterized setting, gradient descent converges linearly to solutions that are bounded away from the boundary as predicted by Theorem 1. Note that in the overparameterized setting, the models consistently converged to solutions with $R \ll R_{teacher}$.

### 5.2 COMPARISON WITH LINEAR ATTENTION MODEL

To isolate the effect of the softmax function, we conduct experiments using a linear attention mechanism. By removing the softmax nonlinearity, we can assess its influence on the behaviour of Gradient Descent. The setup is the same as the one used to produce Figure 1. The only difference is that the model considered here does not contain the Softmax function. More precisely, we use the same teacher dataset and labels considered previously for the softmax attention model represented by the different values of the ratio $R$ in the sparse $R \approx 10^{25}$ and non sparse $R = 10$ case. Figure 3 shows that in the absence of the softmax nonlinearity, we no longer observe the slow convergence of

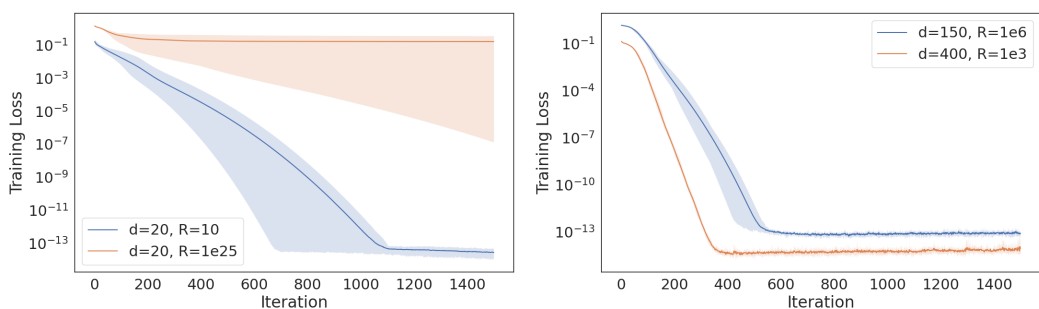

Figure 1: Visualization of the training loss for the model equation 3 on synthetic data for different values of $d$ and the average ratio $R$ of largest to smallest attention probability at the end of training; mean over 10 trials shown with minimum and maximum shaded. In the **overparameterized setting** (**right**), GD does not suffer from poor performance. In the **underparameterized setting** (**left**), the performance of GD depends on the ratio $R$ and is worse when the solutions are closer to the boundary of the simplex.

the loss under GD when the attention is sparse (orange curve). This validates our conclusions about the role of the softmax nonlinearity.

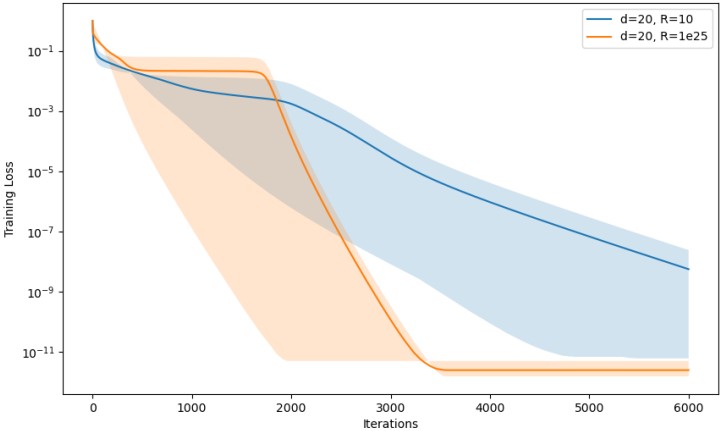

Figure 2: Visualization of the training loss under linear attention (no softmax).

### 5.3 EXPERIMENTS WITH VISION TRANSFORMER (VIT) ON MNIST

Finally, we extend our analysis to more realistic scenarios by training a Vision Transformer (ViT) with 6 layers and 4 heads on the MNIST dataset. We choose a patch size of 4, the embedding dimension that we consider is 64 and the MLP dimension is 256. The labels are once again generated using teacher models that enforce different sparsity levels of the attention scores. The ratio $R$ across layers and heads for the sparse attention setting (orange curve) is $R \approx 10^7$, whereas for the non sparse or less sparse version (blue curve), the average ratio is $R \approx 20$. The model is trained using Stochastic Gradient Descent (SGD) with a constant step size $\eta = 0.01$ that was chosen using a grid search over the step sizes $[0.001, 0.01, 0.1]$. The experiment shows that the conclusions obtained from the theoretical analysis of the simplified model also hold for more practical architectures such as the Vision Transformer : (S)GD suffers from slow convergence as the attention probabilities become sparser.

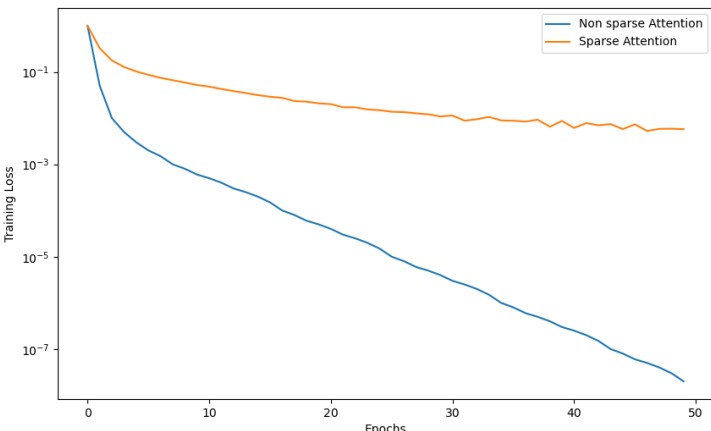

Figure 3: Training loss progression for the Tiny ViT Model averaged over three runs. The observed convergence pattern closely aligns with our theoretical predictions derived from the simplified attention model: the sparsity of the softmax attention leads to slow convergence of GD.

## 6 CONCLUSION

In this paper, we investigated the optimization challenges associated with training attention-based models using gradient descent. Because the softmax function is a feature of attention models that is absent in the core components of other architectures such as CNNs, we focused on the role that it may play in the training dynamics. We established that in overparameterized settings, gradient descent can achieve linear convergence under certain conditions by relying on a local PL and smoothness analysis, since overparameterization implies the existence of non-sparse solutions. We highlighted the limitations of this setting and showed through a local analysis, in the more realistic *underparameterized* setting, how the convergence behavior of gradient descent is affected by the distribution of attention scores. Specifically, as attention probabilities near the boundary of the probability simplex, the optimization problem becomes increasingly ill-conditioned, resulting in slower convergence. Our results highlight the necessity of considering the specific characteristics of the Softmax function and its influence on the local curvature of the loss landscape in the training of attention models. Our work sets the stage for developing more effective training algorithms that can overcome the limitations identified in our analysis. Future work could explore the optimization dynamics of adaptive methods, which have been more successful in the training of transformers, and investigate their ability to better handle the challenges identified in our work, especially in underparameterized regimes.

Finally, we discuss some of the limitations of our work. First, our theoretical results are derived for a simplified model under certain assumptions, which may not fully capture the complexity of real-word data and architectures. In the overparameterized setting, our results required some conditions on the initialization and were limited to strongly convex losses. In the underparameterized setting, our analysis only captures the local dynamics and does not apply globally. Moreover, our focus on the dynamics with a fixed value vector $\mathbf{v}$ and key-query weight matrix $\mathbf{W}$ may not take into account important interactions between these weights and query vectors. These limitations suggest that further research is needed to develop technical tools and frameworks that can handle these complexities. Nonetheless, we believe that our work provides valuable insights and can be a stepping stone towards a better understanding of more complicated models.

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

APPENDIX:

## A    PROOF OF LEMMA 1

**Lemma 1.** *Let $\mathbf{z}$ be a vector in $\mathring{\Delta}^{n-1}$. The matrix $\mathbf{J}(\mathbf{z})$ satisfies the following:*

1. *$\mathbf{J}(\mathbf{z})$ is a symmetric positive semidefinite matrix.*

2. *The vector $\mathbb{1}_n$ is the eigenvector associated with the eigenvalue 0, i.e. $\mathbf{J}(\mathbf{z})\mathbb{1}_n = 0$.*

3. *Let $\lambda_1(\mathbf{z}) \leq \cdots \leq \lambda_n(\mathbf{z})$ denote the eigenvalues of $\mathbf{J}(\mathbf{z})$ and let $\tilde{\mathbf{z}}$ be a vector whose entries are the entries of $\mathbf{z}$ sorted in ascending order $\tilde{\mathbf{z}}_i \leq \tilde{\mathbf{z}}_{i+1}$ for $i \in [n-1]$. The eigenvalues satisfy*

$$0 = \lambda_1(\mathbf{z}) < \tilde{\mathbf{z}}_1 \leq \lambda_2(\mathbf{z}) \leq \cdots \leq \lambda_n(\mathbf{z}) \leq \tilde{\mathbf{z}}_n < 1. \tag{8}$$

*Proof.* Let $\mathbf{z} \in \Delta^{n-1}$. Recall that $\mathbf{J}(\mathbf{z}) = \operatorname{diag}(\mathbf{z}) - \mathbf{z}\mathbf{z}^\top = \operatorname{diag}(\mathbf{z})(I_n - \mathbb{1}_n\mathbf{z}^\top)$.

1. We have $\mathbf{J}(\mathbf{z})^\top = (\operatorname{diag}(\mathbf{z}) - \mathbf{z}\mathbf{z}^\top)^\top = \mathbf{J}(\mathbf{z})$, therefore $\mathbf{J}(\mathbf{z})$ is symmetric. Moreover, $\mathbf{J}(\mathbf{z})$ is the product of a diagonal positive semidefinite matrix and a projection, therefore it is positive semidefinite.

2. $\mathbf{J}(\mathbf{z})\mathbb{1}_n = \operatorname{diag}(\mathbf{z})\mathbf{z} - \mathbf{z}\mathbf{z}^\top\mathbf{z} = 0$.

3. To prove the inequality on the eigenvalues, consider Corollary 4.3.5 in Horn & Johnson (1985) which we restate here:

**Corollary 1.** *Horn & Johnson (1985) Let $A, B \in M_n$ be Hermitian. Suppose that $B$ is singular and* $\operatorname{rank} B = r$. *Then*

$$\lambda_i(A + B) \leq \lambda_{i+r}(A), \quad i = 1, \ldots, n - r \tag{19}$$

By setting $A = \mathbf{J}(\mathbf{z})$ and $B = \mathbf{z}\mathbf{z}^\top$, we have that both matrices are symmetric, $\operatorname{rank}(B) = 1$, and $A + B = \operatorname{diag}(\mathbf{z})$. Therefore

$$\lambda_i(\operatorname{diag}(\mathbf{z})) \leq \lambda_{i+1}(\mathbf{J}(\mathbf{z})) \tag{20}$$

Since the eigenvalues of $\operatorname{diag}(z)$ are $\mathbf{z}_1, \ldots, \mathbf{z}_n$, we just need to order them from smallest to largest and we get

$$\tilde{\mathbf{z}}_1 \leq \lambda_2(\mathbf{z}) \leq \cdots \leq \lambda_n(\mathbf{z}) \leq \tilde{\mathbf{z}}_n \tag{21}$$

Finally, since $\mathbf{z}$ is in the interior, we know that $\tilde{\mathbf{z}}_1 > 0$ and $\tilde{\mathbf{z}}_n < 1$, which concludes our proof.    $\square$

## B    PROOF OF THEOREM 1

**Theorem 1.** *Suppose $d \geq N$, and let $c_1$ and $c_2$ be some positive constants such that $c_1 < 1$ and $c_2 < \min(2, 1/c_1)$. Define the following quantities:*

$$C_\sigma = \max_{1 \leq i \leq N} \sigma_{max}(\mathbf{X}_i), \quad L' = \sigma_{max}^2(W)C_\sigma^3 L\|\mathbf{v}\|\left(2C_\sigma\|\mathbf{v}\| + 3\sqrt{\frac{2(\mathcal{L}_0 - \mathcal{L}^*)}{m}}\right)$$

$$\mu = \frac{c_1}{L'}, \quad \eta = \frac{c_2}{L'}$$

$$C_p = \sqrt{\frac{2(\mathcal{L}_0 - \mathcal{L}^*)}{mN}}\frac{\sigma_{max}(W)C_\sigma^2\|\mathbf{v}\|c_2 L}{L'(1 - \sqrt{1 - c_1 c_2})}, \quad M_p = \|\mathbf{p}_0\| + C_p$$

$$\gamma = \frac{1}{\sqrt{N}}(\min(2, 3C_\sigma\sigma_{max}(\mathbf{W})C_p)C_\sigma^2\sigma_{max}(\mathbf{W})\|\mathbf{v}\|_2)$$

*If $\sigma_{min}(\mathbf{F}(\mathbf{p}_0)) > \gamma + \sqrt{\frac{\mu}{2m}}$, then gradient descent on the function $\mathcal{L}$ with step size $\eta$ converges at a linear rate:*

$$\mathcal{L}(t) - \mathcal{L}^* \leq (1 - \eta\mu)^t(\mathcal{L}(0) - \mathcal{L}^*), \quad \text{for all } t \geq 0. \tag{14}$$

*Moreover, the weights $\mathbf{p}$ and $\mathbf{v}$ remain bounded throughout the trajectory*

$$\|\mathbf{p}_t\|_2 \leq M_p, \quad \text{for all } t \geq 0.$$

*Proof.* We will show by strong induction that for every iteration $t \geq 0$

$$\begin{cases} \|\mathbf{p}_t\|_2 \leq M_p \\ \mathcal{L}(t) - \mathcal{L}^* \leq (1 - \eta\mu)^t(\mathcal{L}(0) - \mathcal{L}^*), & \text{for all } t \geq 0 \end{cases} \tag{22}$$

We start by deriving an upper bound on the gradient of $\mathcal{L}$:

$$\|\nabla_{\mathbf{p}}\mathcal{L}_t\|_2 = \frac{1}{N}\|\sum_{i=1}^{N}\mathbf{K}_i^\top \mathbf{J}(\phi(K_i\mathbf{p}_t))\mathbf{X}_i\mathbf{v}\nabla\ell(\phi(\mathbf{K}_i\mathbf{p}_t)^\top\mathbf{X}_i\mathbf{v}, y_i)\|_2$$

$$\leq \frac{1}{N}\sum_{i=1}^{N}\|\mathbf{K}_i^\top\mathbf{J}(\phi(K_i\mathbf{p}_t))\mathbf{X}_i\mathbf{v}_t\nabla\ell(\phi(\mathbf{K}_i\mathbf{p}_t)^\top\mathbf{X}_i\mathbf{v}_t, y_i)\|_2$$

$$\leq \frac{1}{N}\sum_{i=1}^{N}|\nabla\ell(\phi(\mathbf{K}_i\mathbf{p}_t)^\top\mathbf{X}_i\mathbf{v}, y_i)|\sigma_{max}(\mathbf{X}_i)\sigma_{max}(\mathbf{K}_i)\sigma_{max}(\mathbf{J}(\phi(\mathbf{K}_i\mathbf{p}_t)))\|\mathbf{v}\|_2$$

$$\leq \sqrt{\frac{2(\mathcal{L}_t - \mathcal{L}^*)}{mN}}LC_\sigma^2\sigma_{max}(\mathbf{W})\|\mathbf{v}\|_2 \tag{23}$$

where the second inequality is a result of the triangle inequality and the third inequality uses the sub-multiplicativity of the spectral norm. To obtain the last inequality, we have used the fact that $\sigma_{max}(\mathbf{J}) \leq 1$ as a consequence of Lemma 1.

Now suppose that the inductive hypothesis holds for all $s \in [t]$. We will use it to bound the deviations of the weights from their value at initialization:

$$\|\mathbf{p}_{t+1} - \mathbf{p}_0\|_2 = \|\sum_{s=0}^{t}(\mathbf{p}_{s+1} - \mathbf{p}_s)\|_2$$

$$\leq \sum_{s=0}^{t}\|\mathbf{p}_{s+1} - \mathbf{p}_s\|_2$$

$$= \eta\sum_{s=0}^{t}\|\nabla_{\mathbf{p}}\mathcal{L}_s\|_2$$

$$\leq \eta\sum_{s=0}^{t}\sqrt{\frac{2(\mathcal{L}_s - \mathcal{L}^*)}{mN}}LC_\sigma^2\sigma_{max}(\mathbf{W})\|\mathbf{v}\|_2$$

$$\leq L\eta C_\sigma^2\sigma_{max}(\mathbf{W})\|\mathbf{v}\|_2\sqrt{\frac{2}{mN}}\sum_{s=0}^{t}\sqrt{(1 - \eta\mu)^s(\mathcal{L}_0 - \mathcal{L}^*)}$$

$$\leq L\eta C_\sigma^2\sigma_{max}(\mathbf{W})\|\mathbf{v}\|_2\sqrt{\frac{2(\mathcal{L}_0 - \mathcal{L}^*)}{mN}}\sum_{s=0}^{+\infty}\sqrt{(1 - \eta\mu)^s}$$

$$= \sqrt{\frac{2(\mathcal{L}_0 - \mathcal{L}^*)}{mN}}\frac{L\eta C_\sigma^2\sigma_{max}(\mathbf{W})\|\mathbf{v}\|_2}{1 - \sqrt{1 - \eta\mu}} = C_p \tag{24}$$

Therefore, we have:

$$\|\mathbf{p}_{t+1}\|_2 \leq \|\mathbf{p}_0\|_2 + C_p = M_p \tag{25}$$

Our next step consists in showing local smoothness on the interval $[\mathbf{p}_t, \mathbf{p}_{t+1}]$. We define $\mathbf{p}_{t+\tau} = \mathbf{p}_t + \tau(\theta_{t+1} - \mathbf{p}_t)$, $\tau \in [0, 1]$

$$\|\nabla_{\mathbf{p}}\mathcal{L}(\mathbf{p}_{t+\tau}) - \nabla_{\mathbf{p}}\mathcal{L}(\mathbf{p}_t)\|_2 = \frac{1}{N}\left\|\sum_{i=1}^{N}\mathbf{K}_i^{\top}\left(\mathbf{J}(\phi(\mathbf{K}_i\mathbf{p}_{t+\tau}))\mathbf{X}_i\mathbf{v}\mathbf{g}_i(\mathbf{p}_{t+\tau}) - J(\phi(\mathbf{K}_i\mathbf{p}_t))\mathbf{X}_i\mathbf{v}\mathbf{g}_i(\mathbf{p}_t)\right)\right\|_2$$

$$\leq \frac{1}{N}\sum_{i=1}^{N}\left\|\mathbf{K}_i^{\top}\left(\mathbf{J}(\phi(\mathbf{K}_i\mathbf{p}_{t+\tau}))\mathbf{X}_i\mathbf{v}\mathbf{g}_i(\mathbf{p}_{t+\tau}) - J(\phi(\mathbf{K}_i\mathbf{p}_t))\mathbf{X}_i\mathbf{v}\mathbf{g}_i(\mathbf{p}_t)\right)\right\|_2$$

$$\leq \frac{1}{N}\sum_{i=1}^{N}\|\mathbf{K}_i\|_2\left\|\mathbf{J}(\phi(\mathbf{K}_i\mathbf{p}_{t+\tau}))\mathbf{X}_i\mathbf{v}\mathbf{g}_i(\mathbf{p}_{t+\tau}) - J(\phi(\mathbf{K}_i\mathbf{p}_t))\mathbf{X}_i\mathbf{v}\mathbf{g}_i(\mathbf{p}_t)\right\|_2$$

$$\leq \frac{1}{N}C_{\sigma}\sigma_{max}(\mathbf{W})\sum_{i=1}^{N}\left\|\mathbf{J}(\phi(\mathbf{K}_i\mathbf{p}_{t+\tau}))\mathbf{X}_i\mathbf{v}\mathbf{g}_i(\mathbf{p}_{t+\tau}) - J(\phi(\mathbf{K}_i\mathbf{p}_t))\mathbf{X}_i\mathbf{v}\mathbf{g}_i(\mathbf{p}_t)\right\|_2$$

Next, we bound each term $A_i = \|\mathbf{J}(\phi(\mathbf{K}_i\mathbf{p}_{t+\tau}))\mathbf{X}_i\mathbf{v}\mathbf{g}_i(\mathbf{p}_{t+\tau}) - J(\phi(\mathbf{K}_i\mathbf{p}_t))\mathbf{X}_i\mathbf{v}\mathbf{g}_i(\mathbf{p}_t)\|_2$ in the sum as follows:

$$A_i = \|(\mathbf{J}(\phi(\mathbf{K}_i\mathbf{p}_{t+\tau})) - J(\phi(\mathbf{K}_i\mathbf{p}_t)))\mathbf{X}_i\mathbf{v}\mathbf{g}_i(\mathbf{p}_{t+\tau}) + J(\phi(\mathbf{K}_i\mathbf{p}_t))\mathbf{X}_i\mathbf{v}(\mathbf{g}_i(\mathbf{p}_{t+\tau}) - \mathbf{g}_i(\mathbf{p}_t))\|_2$$

$$\leq \|(\mathbf{J}(\phi(\mathbf{K}_i\mathbf{p}_{t+\tau})) - J(\phi(\mathbf{K}_i\mathbf{p}_t)))\mathbf{X}_i\mathbf{v}\mathbf{g}_i(\mathbf{p}_{t+\tau})\|_2 + \|J(\phi(\mathbf{K}_i\mathbf{p}_t))\mathbf{X}_i\mathbf{v}(\mathbf{g}_i(\mathbf{p}_{t+\tau}) - \mathbf{g}_i(\mathbf{p}_t))\|_2$$

$$\leq C_{\sigma}\|\mathbf{v}\|_2(\|\mathbf{J}(\phi(\mathbf{K}_i\mathbf{p}_{t+\tau})) - J(\phi(\mathbf{K}_i\mathbf{p}_t))\|_2\|\mathbf{g}_i(\mathbf{p}_t)\|_2 + \|J(\phi(\mathbf{K}_i\mathbf{p}_{t+\tau}))\|_2\|\mathbf{g}_i(\mathbf{p}_{t+\tau}) - \mathbf{g}_i(\mathbf{p}_t)\|_2)$$

And we have:

$$\|\mathbf{J}(\phi(\mathbf{K}_i\mathbf{p}_{t+\tau})) - J(\phi(\mathbf{K}_i\mathbf{p}_t))\|_2 \leq 3C_{\sigma}\sigma_{max}(\mathbf{W})\|\mathbf{p}_{t+\tau} - \mathbf{p}_t\|_2, \tag{26}$$

$$\|\mathbf{g}_i(\mathbf{p}_t)\|_2 \leq L\sqrt{\frac{2(\ell(f_{t+\tau}^i) - \ell^*)}{m}}, \quad \text{(Strong convexity and smoothness of } \ell) \tag{27}$$

$$\|J(\phi(\mathbf{K}_i\mathbf{p}_{t+\tau}))\|_2 \leq 1, \quad \text{(Lemma 1)} \tag{28}$$

$$\|\mathbf{g}_i(\mathbf{p}_{t+\tau}) - \mathbf{g}_i(\mathbf{p}_t)\|_2 \leq L\|f_{t+\tau}^i - f_t^i\|_2, \quad \text{(Lipschitz continuity of the gradient of } \ell)$$
$$\leq LC_{\sigma}\|\mathbf{v}\|_2\|\phi(\mathbf{K}_i\mathbf{p}_{t+\tau}) - \phi(\mathbf{K}_i\mathbf{p}_t)\|_2$$
$$\leq 2LC_{\sigma}^2\sigma(\mathbf{W})\|\mathbf{v}\|_2\|\mathbf{p}_{t+\tau} - \mathbf{p}_t\|_2. \tag{29}$$

We can conclude that

$$\|\nabla_{\mathbf{p}}\mathcal{L}(\mathbf{p}_{t+\tau}) - \nabla_{\mathbf{p}}\mathcal{L}(\mathbf{p}_t)\|_2 \leq C_{\sigma}^2\sigma(\mathbf{W})\|\mathbf{v}\|_2\left(2LC_{\sigma}^2\sigma(\mathbf{W})\|\mathbf{v}\|_2 + 3C_{\sigma}\sigma(\mathbf{W})L\sum_{i=1}^{N}\sqrt{\frac{2\ell(f_{t+\tau}^i) - \ell^*}{mN^2}}\right)\|\mathbf{p}_{t+\tau} - \mathbf{p}_t\|_2$$

$$\leq C_{\sigma}^3\sigma^2(\mathbf{W})\|\mathbf{v}\|_2\left(2LC_{\sigma}\|\mathbf{v}\|_2 + 3L\sqrt{\frac{2(\mathcal{L}_0 - \mathcal{L}^*)}{m}}\right)\|\mathbf{p}_{t+\tau} - \mathbf{p}_t\|_2$$

$$= L'\|\mathbf{p}_{t+\tau} - \mathbf{p}_t\|_2 \tag{30}$$

Finally, we determine a uniform lower bound for $\sigma_{min}(\mathbf{F}(\mathbf{p}_t))$ to establish the PL inequality.

We consider the deviation of $\mathbf{F}(\mathbf{p}_t)$ from its initialization:

$$\|\mathbf{F}(\mathbf{p}_t) - \mathbf{F}(\mathbf{p}_0)\|_F^2 = \frac{1}{N^2}\sum_{i=1}^{N}\|\mathbf{K}_i^{\top}\mathbf{J}(\phi(\mathbf{K}_i\mathbf{p}_t))\mathbf{X}_i\mathbf{v} - \mathbf{K}_i^{\top}\mathbf{J}(\phi(\mathbf{K}_i\mathbf{p}_0))\mathbf{X}_i\mathbf{v}\|_2^2$$

$$\leq \frac{C_{\sigma}^4\sigma_{max}^2(\mathbf{W})\|\mathbf{v}\|_2}{N^2}\sum_{i=1}^{N}\|\mathbf{J}(\phi(\mathbf{K}_i\mathbf{p}_t)) - \mathbf{J}(\phi(\mathbf{K}_i\mathbf{p}_0))\|_2^2$$

Each term in the sum has the following bound:

$$\|\mathbf{J}(\phi(\mathbf{K}_i\mathbf{p}_t)) - \mathbf{J}(\phi(\mathbf{K}_i\mathbf{p}_0))\|_2 \leq 3C_\sigma\sigma_{max}(\mathbf{W})\|\mathbf{p}_t - \mathbf{p}_0\|_2$$
$$\leq \min(2, 3C_\sigma\sigma_{max}(\mathbf{W})C_p)$$

Let $\gamma = \frac{1}{\sqrt{N}}(\min(2, 3C_\sigma\sigma_{max}(\mathbf{W})C_p)C_\sigma^2\sigma_{max}(\mathbf{W})\|\mathbf{v}\|_2)$, we have

$$\|\mathbf{F}(\mathbf{p}_t) - \mathbf{F}(\mathbf{p}_0)\|_F \leq \gamma.$$

Using Weyl's inequality, we conclude that

$$\sigma_{min}(\mathbf{F}(\mathbf{p}_t)) \geq \sigma_{min}(\mathbf{F}(\mathbf{p}_0)) - \gamma > \sqrt{\mu/2m}. \tag{31}$$

Therefore, since the step size satisfies $\eta < \frac{2}{L'}$, we have

$$\mathcal{L}_{t+1} - \mathcal{L}^* \leq \mathcal{L}_t - \frac{\eta}{2}\|\nabla_\theta\mathcal{L}_t\|_2^2 - \mathcal{L}^*$$
$$\leq (1 - \eta\mu)(\mathcal{L}_t - \mathcal{L}^*)$$
$$\leq (1 - \eta\mu)^{t+1}(\mathcal{L}_0 - \mathcal{L}^*). \tag{32}$$

which concludes our proof. $\qquad\square$

## C    PROOF OF PROPOSITION 1

First, we analyze the bound $\gamma + \sqrt{\frac{\mu}{2m}}$ as a function of $\|\mathbf{v}\|_2 = \alpha$.

For very large $\alpha$, we have $L' = C_1\alpha^2 + C_2\alpha$, therefore $\mu = O(1/\alpha^2)$ and $C_p = O(1/\alpha^2)$. Moreover, when $C_p$ is sufficiently small, we have $3C_\sigma\sigma(\mathbf{W})C_p \leq 2$ and $\gamma = O(1/\alpha)$.

Next, we analyze $\sigma_{min}(\mathbf{F}(\mathbf{p}_0))$. We start by showing that $\sigma_{min}(\mathbf{F}(\mathbf{p}_0)) > 0$, then we show that $\sigma_{min}(\mathbf{F}(\mathbf{p}_0)) = O(\alpha)$ to conclude that when $\alpha$ is large enough, the condition is satisfied.

- First, we show that all rows of $\mathbf{F}(\mathbf{p}_0)$ are non zero. From assumptions 4 and 3, $\mathbf{W}$ has full rank and that $\mathbf{X}_i$ is surjective for all $i = 1, \ldots, N$. As a result, $X_i^T$ is injective for all $i$ and

$$\mathbf{J}(\phi(\mathbf{K}_i\mathbf{p}_0))\mathbf{X}_i\mathbf{v} \neq 0 \implies \mathbf{K}_i^\top\mathbf{J}(\phi(\mathbf{K}_i\mathbf{p}_0)\mathbf{X}_i\mathbf{v} \neq 0. \tag{33}$$

  Moreover, we know that $\ker\mathbf{J}(\phi(\mathbf{K}_i\mathbf{p}_0)) = \text{span}(\mathbb{1}_n)$, therefore for all $i = 1, \ldots, N$, we need to ensure that $X_i\mathbf{v} \notin \ker\mathbf{J}(\phi(\mathbf{K}_i\mathbf{p}_0))$. However, if we assume that the $X_i$'s are drawn from a distribution $\mathcal{D}$, then the preimage of $\ker\mathbf{J}(\phi(\mathbf{K}_i\mathbf{p}_0))$ in $\mathbb{R}^d$ has measure 0. Therefore, any choice of vector $\mathbf{v}$ will satisfy the condition with probability 1.

- Assumption 2 guarantees the orthogonality of the rows $\mathbf{F}(\mathbf{p}_0)$, thus ensuring that they are all linearly independent.

- We can write

$$\mathbf{F}(\mathbf{p}_0) = \frac{\alpha}{N}\begin{bmatrix} (\mathbf{K}_1^\top\mathbf{J}(\phi(\mathbf{K}_1\mathbf{p}_t))\mathbf{X}_1\mathbf{u})^\top \\ \vdots \\ (\mathbf{K}_N^\top\mathbf{J}(\phi(\mathbf{K}_N\mathbf{p}_t))\mathbf{X}_N\mathbf{u})^\top \end{bmatrix} \tag{34}$$

  therefore $\sigma_{min}(\mathbf{F}(\mathbf{p}_0)) = O(\alpha)$, which allows us to conclude that for $\alpha$ large enough, we have $\sigma_{min}(\mathbf{F}(\mathbf{p}_0)) > \gamma + \sqrt{\frac{\mu}{2m}}$ with probability 1.

## D    PROOF OF THEOREM 2

**Theorem 2.** *Let $\mathbf{z}^*$ be a stationary point of the dynamics described in equation 16. Suppose that $\mathbf{z}^*$ is in the interior of the simplex, then the following holds:*

*1. The linearization of the system around $\mathbf{z}^*$ is given by $\zeta_{t+1} = \zeta_t - \frac{\eta}{N}\mathbf{J}^2(\mathbf{z}^*)\nabla^2\tilde{\ell}(\mathbf{z}^*)\zeta_t$.*

2. *Let $\mu = \lambda_{min}(\nabla^2 \tilde{\ell}(\mathbf{z}^*))$ and $L = \lambda_{max}(\nabla^2 \tilde{\ell}(\mathbf{z}^*))$. Suppose that $\mu > 0$ and let $\kappa$ be the condition number of $\mathbf{J}^2(\mathbf{z}^*)\nabla^2 \tilde{\ell}(\mathbf{z}^*)$ when restricted to the tangent space of the simplex $T\Delta^{n-1} = \{\mathbf{u} \in \mathbb{R}^n : \mathbb{1}^\top \mathbf{u} = 0\}$. Then we have the following lower bound on $\kappa$*

$$\kappa \geq \frac{\mu \max_i (\mathbf{z}_i^*)^2}{L \min_i (\mathbf{z}_i^*)^2}. \tag{18}$$

*Proof.* 1. To linearize the system around $\mathbf{z}^*$, we write the dynamics of $\zeta_t = \mathbf{z}_t - \mathbf{z}^*$:

$$\zeta_{t+1} = \zeta_t - \frac{\eta}{N}\mathbf{J}^2(\mathbf{z}^* + \zeta_t)\nabla\tilde{\ell}(\mathbf{z}^* + \zeta_t) + \frac{\eta^2}{2N^2}D^2\phi(\mathbf{c}_t)(\mathbf{J}(\mathbf{z}^* + \zeta_t)\nabla\tilde{\ell}(\mathbf{z}^* + \zeta_t), \mathbf{J}(\mathbf{z}^* + \zeta_t)\nabla\tilde{\ell}(\mathbf{z}^* + \zeta_t))$$

Next we analyze the second and third terms. Let $\mathbf{V}(\mathbf{x}) = \mathbf{J}(\mathbf{x})f(\mathbf{x})$ and $f(\mathbf{x}) = J(\mathbf{x})\nabla\tilde{\ell}(\mathbf{x})$. The linearization of $\mathbf{V}(\mathbf{z}^* + \zeta_t)$ can be expressed as:

$$\mathbf{V}(\mathbf{z}^* + \zeta_t) = \mathbf{V}(\mathbf{z}^*) + DV(\mathbf{z}^*)\zeta_t = DV(\mathbf{z}^*)\zeta_t \tag{35}$$

Where we have used the fact that $\mathbf{V}(\mathbf{z}^*)$ because $\mathbf{z}^*$ is a stationary point.

To compute $D\mathbf{V}(\mathbf{x})$, we use the product rule componentwise. Recall that for two vector-valued functions $H_1 : \mathbb{R}^n \to \mathbb{R}^m$ and $H_2 : \mathbb{R}^n \to \mathbb{R}^m$, the componentwise product is defined as $H_1 \odot H_2(\mathbf{x}) = \text{diag}(H_1(\mathbf{x}))H_2(\mathbf{x}) + \text{diag}(H_1(\mathbf{x}))H_2(\mathbf{x})$. And the product rule yields:

$$D(H_1 \odot H_2)(\mathbf{x}) = \text{diag}(H_1(\mathbf{x}))DH_2(\mathbf{x}) + \text{diag}(H_1(\mathbf{x}))DH_2(\mathbf{x})$$

In our case, we have $\mathbf{V}_i(\mathbf{x}) = \mathbf{x}_i\hat{f}_i(\mathbf{x})$ with $\hat{f}_i(\mathbf{x}) = f_i(\mathbf{x}) - \mathbb{1}\bar{f}(\mathbf{x})$ and $\bar{f}(\mathbf{x}) = \mathbf{x}^\top f(\mathbf{x})$

And we have

$$D\hat{f}(\mathbf{x}) = Df(\mathbf{x}) - \mathbb{1}(\mathbf{x}^\top Df(\mathbf{x}) + f(\mathbf{x})^\top) = (I - \mathbb{1}\mathbf{x}^\top)Df(\mathbf{x}) - \mathbb{1}f(\mathbf{x})^\top$$

As a result:

$$D\mathbf{V}(\mathbf{x}) = D(\text{diag}(\mathbf{x})\hat{f}(\mathbf{x})) \tag{36}$$
$$= \text{diag}(\mathbf{x})D\hat{f}(\mathbf{x}) + \text{diag}(\hat{f}(\mathbf{x})) \tag{37}$$
$$= \text{diag}(\mathbf{x})((I - \mathbb{1}\mathbf{x}^\top)Df(\mathbf{x}) - \mathbb{1}f(\mathbf{x})^\top) + \text{diag}(\hat{f}(\mathbf{x})) \tag{38}$$
$$= \mathbf{J}(\mathbf{x})Df(\mathbf{x}) - \mathbf{x}f(\mathbf{x})^\top + \text{diag}(\hat{f}(\mathbf{x})) \tag{39}$$

Let $\mathbf{z}^* \in \mathring{\Delta}^{n-1}$ be a stationary point, we have $f(\mathbf{z}^*) = \mathbf{J}(\mathbf{z}^*)\nabla\ell(\mathbf{z}^*) = 0$ and $\hat{f}(\mathbf{z}^*) = 0$, therefore, we have

$$D\mathbf{V}(\mathbf{z}^*) = \mathbf{J}(\mathbf{z}^*)Df(\mathbf{z}^*)$$

Now we write $D\mathbf{V}(\mathbf{z}^*)$ as a function of $\mathbf{J}$ and $\nabla^2\tilde{\ell}$.

Since $f(\mathbf{x}) = \mathbf{J}(\mathbf{x})F(\mathbf{x})$ with $F(\mathbf{x}) = -\nabla\tilde{\ell}(\mathbf{x})$, then using the same result derived for $D\mathbf{V}(\mathbf{x})$, we obtain

$$Df(\mathbf{x}) = \mathbf{J}(\mathbf{x})DF(\mathbf{x}) - \mathbf{x}F(\mathbf{x})^\top + \text{diag}(\hat{F}(\mathbf{x})).$$

Once again, since $\mathbf{z}^*$ is an interior global minimum, we have $\hat{F}(\mathbf{z}^*) = 0$. Moreover, $F(\mathbf{z}^*)$ is a constant vector. We can thus consider the orthogonal projection onto the tangent space of the simplex $P = I - \frac{1}{n}\mathbb{1}\mathbb{1}^\top$ as follows

$$Df(\mathbf{z}^*)P = \mathbf{J}(\mathbf{z}^*)DF(\mathbf{z}^*)P = -\mathbf{J}(\mathbf{z}^*)\nabla^2\tilde{\ell}(\mathbf{z}^*)P$$

Finally, we conclude that $D\mathbf{V}(\mathbf{z}^*)P = -\mathbf{J}^2(\mathbf{z}^*)\nabla^2\tilde{\ell}(\mathbf{z}^*)P$.

For the third term, we first consider the part $\mathbf{J}(\mathbf{z}^* + \zeta_t)\nabla\tilde{\ell}(\mathbf{z}^* + \zeta_t)$ which we have already analyzed above. Its linearization is:

$$\mathbf{J}(\mathbf{z}^* + \zeta_t)\nabla\tilde{\ell}(\mathbf{z}^* + \zeta_t) \approx \mathbf{J}(\mathbf{z}^*)\nabla\tilde{\ell}(\mathbf{z}^*) + \mathbf{J}(\mathbf{z}^*)\nabla^2\tilde{\ell}(\mathbf{z}^*)\zeta_t = \mathbf{J}(\mathbf{z}^*)\nabla^2\tilde{\ell}(\mathbf{z}^*)\zeta_t. \tag{40}$$

where once again the fact that $\mathbf{z}^*$ is an interior stationary point guarantees that $\mathbf{J}(\mathbf{z}^*)\nabla\tilde{\ell}(\mathbf{z}^*)$.

Moreover, computing the second order derivatives of the Softmax yields:

$$\frac{\partial^2 \phi_i}{\partial \mathbf{x}_k \partial \mathbf{x}_j} = \phi_i(\delta_{ij} - \phi_j)(\delta_{ik} - \phi_k) - \phi_i\phi_k(\delta_{kj} - \phi_j) \in [-2, 2] \tag{41}$$

Since all the terms of the second derivative are bounded, the term $D^2\phi(\mathbf{c}_t)(\mathbf{J}(\mathbf{z}^* + \zeta_t)\nabla\tilde{\ell}(\mathbf{z}^* + \zeta_t), \mathbf{J}(\mathbf{z}^* + \zeta_t)\nabla\tilde{\ell}(\mathbf{z}^* + \zeta_t))$ is quadratic in $\zeta_t$. And we can conclude that the linearization of the gradient descent dynamics on $\mathbf{z}$ is given by:

$$\zeta_{t+1} = \zeta_t - \frac{\eta}{N}\mathbf{J}^2(\mathbf{z}^*)\nabla^2\tilde{\ell}(\mathbf{z}^*)\zeta_t. \tag{42}$$

2. Let $\mu = \lambda_{min}(\nabla^2\tilde{\ell}(\mathbf{z}^*))$ and $L = \lambda_{max}(\nabla^2\tilde{\ell}(\mathbf{z}^*))$. For any $\mathbf{u} \in T\Delta^{n-1}$, we have using the eigendecomposition of $\mathbf{J}(\mathbf{z}^*)$ that

$$\lambda_2(\mathbf{J}(\mathbf{z}^*))\|\mathbf{u}\|_2 \le \|\mathbf{J}(\mathbf{z}^*)\mathbf{u}\|_2 \le \lambda_n(\mathbf{J}(\mathbf{z}^*))\|\mathbf{z}\|_2 \tag{43}$$

where the equalities are achieved when $\mathbf{u}$ is an eigenvector associated with the eigenvalue $\lambda_2(\mathbf{J}^2(\mathbf{z}^*))$ or $\lambda_n(\mathbf{J}^2(\mathbf{z}^*))$.

Therefore, when restricted to the tangent space of the simplex, the eigenvalues of the map $\mathbf{J}^2(\mathbf{z}^*)$ satisfy:

$$\lambda_{max}(\mathbf{J}^2(\mathbf{z}^*)) \le (\max_i(\mathbf{z}_i^*))^2, \quad \text{and} \quad \lambda_{min}(\mathbf{J}^2(\mathbf{z}^*)) \ge (\min_i(\mathbf{z}_i^*))^2 \tag{44}$$

where we have used the results of Lemma 1.

Moreover, we know that

$$\lambda_{max}(\mathbf{J}^2(\mathbf{z}^*)\nabla^2\tilde{\ell}(\mathbf{z}^*)) \ge \lambda_{min}(\mathbf{J}^2(\mathbf{z}^*))\lambda_{max}(\nabla^2\tilde{\ell}(\mathbf{z}^*)) \tag{45}$$

and

$$\lambda_{min}(\mathbf{J}^2(\mathbf{z}^*)\nabla^2\tilde{\ell}(\mathbf{z}^*)) \le \lambda_{max}(\mathbf{J}^2(\mathbf{z}^*))\lambda_{min}(\nabla^2\tilde{\ell}(\mathbf{z}^*)) \tag{46}$$

And we can deduce that

$$\lambda_{max}(\mathbf{J}^2(\mathbf{z}^*)\nabla^2\tilde{\ell}(\mathbf{z}^*)) \ge L(\min_i(\mathbf{z}_i^*))^2$$

$$\lambda_{min}(\mathbf{J}^2(\mathbf{z}^*)\nabla^2\tilde{\ell}(\mathbf{z}^*)) \le \mu(\max_i(\mathbf{z}_i^*))^2 \tag{47}$$

We can thus recover the lower bound on the condition number and conclude our proof.

$\square$

# E  VISUALIZING ATTENTION MAPS FOR VIT

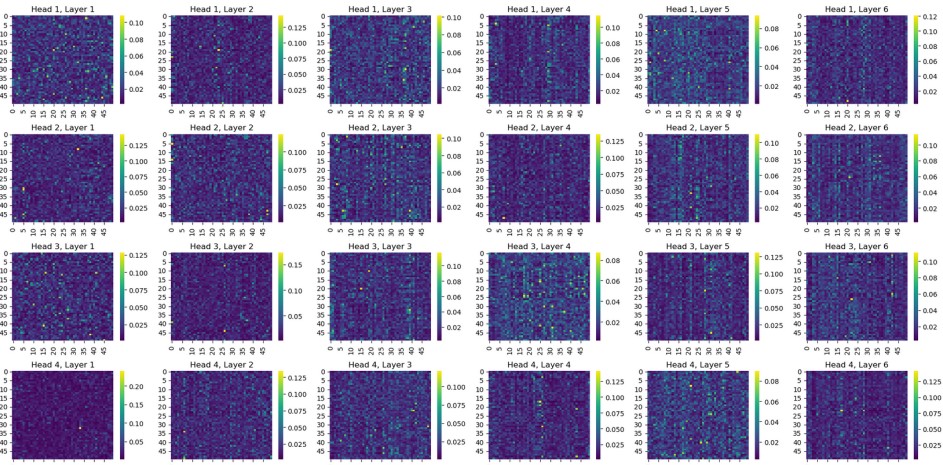

Figure 4: Attention maps obtained at the end of training of the model with $R \approx 20$.

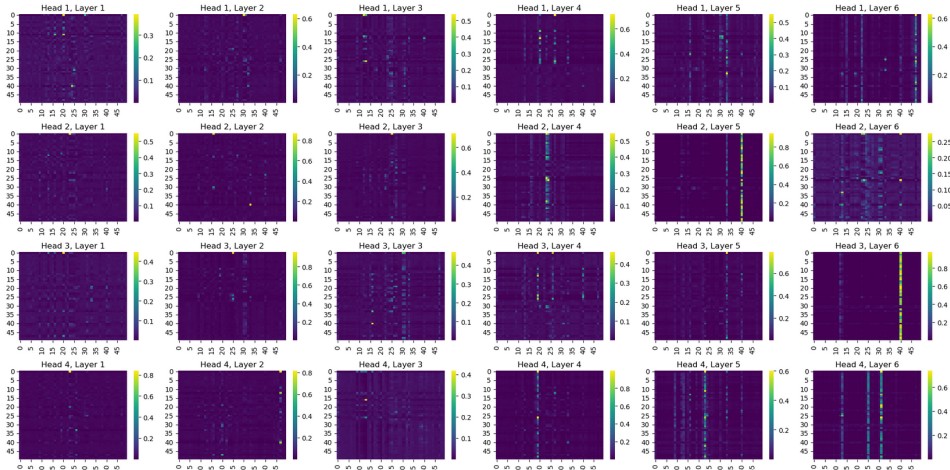

Figure 5: Attention maps obtained at the end of training of the model with $R \approx 10^7$.

# F  BROADER IMPACT

This work provides valuable insights into the optimization challenges associated with training attention-based models, which have become ubiquitous in modern machine learning applications. It could lead to more efficient and effective optimization algorithms for transformers and similar applications, which would be of high interest to the ML community. We do not expect any negative societal bias from this work.

