# OpenReview forum: "Gradient Descent and Attention Models: Challenges Posed by the Softmax Function"
_ICLR.cc/2025/Conference — ICLR 2025 Conference Withdrawn Submission_

### Official Review · Reviewer_Go9x · 2024-10-25

**Soundness:** 2
**Presentation:** 2
**Contribution:** 2
**Rating:** 5
**Confidence:** 3

**Summary:**

This paper aims to explain the challenges gradient descent (GD) faces in efficiently training transformers. The authors provide a theoretical analysis using simplified transformers based on tunable tokens, while keeping the attention weights fixed. In the overparameterized case, they demonstrate that the PL condition holds, leading to a linear convergence rate. In the underparameterized case, they show that when the stationary points are near the boundary of simplex, the gradient condition number increases, leading to ill-conditioning.

**Strengths:**

The paper offers novel insights into why GD struggles with softmax attention, particularly through a local analysis, which provides a fresh perspective.

**Weaknesses:**

1. The setting is somewhat limited. In the simplified transformer model, only the tunable tokens are learned, while the attention matrix remains fixed. This approach may primarily address the challenges GD encounters in prompt-tuning tasks on transformers but does not extend to other tasks, such as language modeling. Therefore, I believe this simplified model does not fully capture the key features of transformers that make GD optimization challenging in practical applications.
2. In Assumption 2, it seems necessary to have  $d \geq n \cdot N$ . Yet, in the underparameterized case where $d \ll N$ , your analysis still relies on Assumption 2. This appears contradictory, and as a result, the equation on line 345 may not hold as stated.
3. The proof of Theorem 2 appears inconsistent with its statement. In Eq. (47), you have  $\lambda_{\text{max}} \geq L(\min z_i)^2$  and  $\lambda_{\text{min}} \leq \mu(\max z_i)^2$, which suggests  $\kappa \geq \frac{L(\min z_i)^2}{\mu(\max z_i)^2}$ . This would lower the  bound, which seems to contradict your argument regarding  $R$. Are there any potential typos?

**Questions:**

1. I wonder if the p-parameterization accurately reflects the true challenges encountered by GD. Have you observed similar issues with the query-key parameterization? Furthermore, in the p-parameterization, you can adapt sparsity by adjusting the variance of p. However, I am curious how the sparsity of attention can be quantified or adapted in the query-key parameterization?
2. Does the paper suggest that GD fails due to the sparse softmax pattern of the target function?
3. The writing could benefit from further refinement:
  - Could you clarify the specific assumptions required for Theorems 1 and 2?
  - Could you elaborate on Assumption 3 and Remark 4? For instance, what is meant by “surjective,” and what do you mean by “In the case of multiple heads, the matrix W has low rank by construction”?
  - The citation of Liu et al. on line 94 seems incorrect; it should refer to [1].
  - The citations in lines 333-334 should use \citep formatting.


[1] Li et al., “A theoretical understanding of shallow vision transformers: Learning, generalization, and sample complexity”, 2023.

---

### Official Review · Reviewer_Q4zA · 2024-11-02

**Soundness:** 3
**Presentation:** 3
**Contribution:** 3
**Rating:** 3
**Confidence:** 3

**Summary:**

This paper analyzes the dynamic characteristics of gradient descent in the attention model with softmax and analyzes the convergence in two cases: over-parameterization and under-parameterization.

**Strengths:**

This result covers both over-parameterized and under-parameterized scenarios, providing a comprehensive understanding of gradient descent's performance in the Softmax attention model.

This study identifies pathologies in the Softmax mechanism, especially the negative impact on convergence speed when attention is sparse.

**Weaknesses:**

1: This paper studies the convergence in an under-parameterized regime, but the analysis mainly focuses on the local convergence near the stationary point.

2: The paper's assumptions are relatively strict (assumption 2, PL conditions, and smooth conditions).

3: Some settings simplify the difficulty of analysis (assumption 1).

**Questions:**

This paper analyzes GD in the Softmax attention model, but in Adam, which is more commonly used in practice, what will these results become, and will it change the impact of Softmax?

---

### Official Review · Reviewer_dDmB · 2024-11-03

**Soundness:** 2
**Presentation:** 3
**Contribution:** 2
**Rating:** 3
**Confidence:** 3

**Summary:**

The paper studies convergence behavior of GD in optimizing attention. The motivation that the authors lay out is the empirically observed poor performance of GD, in constant to Adam, in transformer models (unlike say CNNs). Specifically the authors study prompt-attention in a simplified setting where data are assumed orthogonal and only the prompt vector is optimized (linear head is fixed). For this, they split the analysis into what they call overparameterized (OP) and underparameterized (UP) regimes depending on whether the parameters d exceed the number of samples N or not. In OP, they show that if loss function is strongly convex, then under appropriate initialization, the parameter norm remains bounded and convergence is linear. However, they argue this setting is restrictive because parameter norm remaining bounding cannot promote sparse solutions. Moreover, they argue that d>N is not practical. Thus, turning focus to the UP, they study local dynamics (specifically a linearization of attention vector updates around a stationary point) and show that these dynamics are to the first-order governed by a system with condition number that scales proportional to the ratio of largest to smallest attention probabilities (at the stationary point). This, they say, suggests that optimization slows down more and more if the stationary point is such that attention probabilities are on the boundary of the simplex. Some experiments on synthetic data and an experiment on a small ViT transformer trained on MNIST data are used to justify the theory.

**Strengths:**

Strengths:
1. study of attention optimization dynamics is important topic. The motivation of particularly understanding why GD does not perform well, while Adam does is certainly a relevant topic in the community

2. the findings seem to suggest that GD on attn models performs particularly poorly when the attention converges to sparse probability vectors: this is an interesting point to highlight

3. paper is generally easy to follow and quite well written/organized

**Weaknesses:**

1. Unfortunately, after a very promising introduction and motivation on performance of GD vs Adam on transformers, I found the results somewhat underwhelming. I would hope that at least in this very simplified (not easy, but admittedly simplified with many assumptions to help analysis), the authors would present a setting where Adam>GD. This is not even given in the experiments.  The main theoretical results are also somewhat limited. (For OP, it is rather well understood that sufficient overparameterization and appropriate initialization induces some form of PL with square loss. For UP, if I understand correct, the local analysis is essentially studying conditioning of a linearization of a quadratic approximation of GD updates of attention outputs; e.g. the quadratic approximation in (16) seems to be ‘hiding under the rug’ the fact that optimized parameters p_t need to diverge for z_t to end up being sparse.)

2. As mentioned above, some assumptions made are rather strong. In particular, the orthogonality assumption and the strong convexity assumptions. I ‘d prefer if the authors simply acknowledge that they need these for analysis and also provide some insights on where they are critical and how one could potentially relax them rather than attempting to justify them as reasonable (eg see Remark 1)

3. The experiments are rather limited. Even judging this as a theory paper, I d expect some more results to be presented (synthetic OK), e.g. (i) Adam performance for this synthetic model, (ii) estimation error (together with optimization error) since you are considering a student-teacher setup, (iii) another  setting where sparsity is more explicitly imposed and desired at the output, (iv) other loss functions: does the same behavior hold for logistic loss, joint training of v and p

**Questions:**

1. Ass 2 is clearly strong. Appeal to Wu et al is not sufficient to justify. At least I would hope the authors give some examples where this might be (approximately) satisfied. In fact, note that even the assumption d\geq n is not satisfied in most cases since the context window is typically larger than d. Q: Sorry if I miss something, but doesn’t ass 2  generally require dn>N for orthogonality to hold? (Thinking of flattened versions of K_i of dimension dn each and you need N such vectors to be pairwise orthogonal)
 Overall, as I mentioned: I find the connection of GD speed to sparsity interesting, but I feel this is not explicit in the results but rather seems to be a ‘heuristic’ interpretation of the results

2. the loss strong convexity should be stated explicitly in thm 1

3. What is the point of studying d>N since as the authors argue (I) attention probabilities can’t get sparse to select relevant tokens (ii) d<N in practice

4. I am a bit confused with the distinction between OP and UP in the following sense: shouldn’t it be that in general loss can’t be 0 in the UP regime? If so, then how is loss reaching zero in fig 1 left? Is it true that if d>N then loss can be driven to zero, otherwise not necessarily so?

5. why theorem 2 does not hold in the OP regime?

6.  could you please clarify whether thm 2 needs strong convexity of \ell? If not, then what is \mu? Could you please give some examples

7. For the synthetic experiments please see my comments on limitations. e.g. Can you also plot ||p_t - p_teacher||? And, does Adam resolve the issue in this simple model

8. Sec. 5.3: could you please clarify: since you are training with MNIST what is the role of the teacher model? Aren’t labels predetermined?

9. I believe there is quite a few missing references on recent works on optimization dynamics of attention (other than the few surveyed in the related work section).  Vasudeva et al.’s ‘Implicit bias and fast convergence rates for self-attention’ seems particularly related as they show adaptive step size in the form of normalized GD leads to faster loss convergence in simplified attention. Does for example normalized GD resolve the issue that you find GD has?

10.  typos: X_i s should be capitalized in the text surrounding Eq (33)

11.  should “u” be a “v” in Eq (34)? If so, does this cause any problems with the conclusion that sigma_min is O(\alpha)

---

### Official Review · Reviewer_Pywq · 2024-11-04

**Soundness:** 3
**Presentation:** 3
**Contribution:** 1
**Rating:** 3
**Confidence:** 4

**Summary:**

This work analyzes the dynamics of gradient descent training for a single-head attention model. It shows that softmax attention causes the local curvature of the loss landscape to be ill-conditioned and hampers the convergence rate.

**Strengths:**

The attention mechanism is a core component of transformers and understanding and improving its optimization dynamics is an important area of research.

**Weaknesses:**

There are several missing references in the discussion on related work, e.g. [1-5]. The authors don’t cite [1], due to which some of the statements about the contribution seem misleading. [1] analyzes gradient-based methods with adaptive step size and shows global convergence and finite-time fast convergence rates for parameter convergence for a single-head self-attention model. In the Introduction, the authors state that prior works lack convergence rates, and in the Conclusion, they even mention analyzing adaptive step sizes as a direction for future work.

While the observation that using linear attention instead of softmax attention can speed up convergence is somewhat interesting, it is not very insightful given that [1] already shows that using adaptive step size provably speeds up convergence compared to gradient descent with fixed step size.

The analysis is done under very strong assumptions on the model. The authors consider prompt attention with a fixed linear decoder, instead of the commonly used self-attention. Since the decoder weights are fixed, the loss only decreases in a small range as the attention weights change and it does not converge to 0 in this setting. So, deriving a convergence rate in this setting is not very informative.

The experimental section is also very weak, the authors should include results with more complex models and datasets.

References:

[1] Bhavya Vasudeva, Puneesh Deora, and Christos Thrampoulidis. Implicit bias and fast convergence rates for self-attention, 2024.

[2] Puneesh Deora, Rouzbeh Ghaderi, Hossein Taheri, Christos Thrampoulidis. On the Optimization and Generalization of Multi-head Attention, TMLR 2024.

[3] Yuandong Tian, Yiping Wang, Beidi Chen, and Simon Du. Scan and snap: Understanding training dynamics and token composition in 1-layer transformer, NeurIPS 2023.

[4] Yuandong Tian, Yiping Wang, Zhenyu Zhang, Beidi Chen, and Simon Du. Joma: Demystifying multilayer transformers via joint dynamics of mlp and attention, ICLR 2024.

[5] Heejune Sheen, Siyu Chen, Tianhao Wang, and Harrison H. Zhou. Implicit regularization of gradient flow on one-layer softmax attention, 2024.

**Questions:**

Can the authors justify their contribution compared to [1]? It would be interesting if the analysis could be extended to joint training or if the benefit of linear attention over softmax can be shown theoretically.

Can the authors enhance the experimental section? Why is Fig. 2 missing the other setting from Fig. 1? The size of Figs. 2 and 3 should be reduced.

---

### Official Review · Reviewer_cKXc · 2024-11-09

**Soundness:** 2
**Presentation:** 2
**Contribution:** 2
**Rating:** 5
**Confidence:** 3

**Summary:**

This work aims to explain the reasons for the poor performance of SGD when training Transformer models, by showing that softmax used in the Attention module results in a condition number that scales in proportion to the square of the ratio of the largest to the smallest attention probabilities, and thus decelerating convergence to sparse attention matrices. The theoretical analysis is carried out in the confined setting of prompt tuning, and convergence rates are studied under both over-parameterized and under-parameterized regimes (defined as per model dimension d > or < number of samples N). The former is unlikely to hold in practice, and the latter is where the result on the condition number is shown. The analysis is accompanied by experiments in synthetic or small settings that illustrate the core claim.

**Strengths:**

- Interesting theoretical analysis on the role of softmax and how it impacts convergence. The analysis is presented in both over and under parameterized scenarios, the latter of which shows the dependence on the maximum and minimum attention probability.

- The paper presents some toy experiments which further back these claims.

- The takeaway is insightful and might inform future studies.

**Weaknesses:**

### Theoretical Limitations

(Assumption 1)

- Oversimplified Setup: The theoretical setup of prompt tuning might oversimplify the complex reasons why SGD struggles with Transformers. The softmax issue, while potentially significant, may not be the sole culprit.

- Ignoring Transformer Heterogeneity: Transformers are known for their heterogeneous curvature landscapes [1, 2]. Parameterizing query-key matrices as a single matrix can further alter this landscape [2].

I think it would be beneficial to compare how crucial each of these factors are for hindering the use of SGD with Transformers, and a discussion on these recent work would be helpful.


(Assumption 2)
- If what is being assumed is that for each sequence the token embeddings for distinct tokens are orthogonal, this could be just written as X X^\top = I. Why involve it together with key weights? And this is also used as such in lines 345-348, which would imply that this assumption is misleading, and it is beyond just the input data. Besides, I think this assumption is also quite drastic as there is no reason to assume that the tokens in a sentence are 'orthogonal'. This is also assumed for every sequence, if I understand correctly?

### Methodological Limitations

- How so adaptive methods bypass this issue caused by Softmax? Can the authors comment how AdamW ends up combating this issue? Otherwise this would be an incomplete explanation.

- The focus here is on a local analysis, but the problems with SGD start appearing from the beginning of training language models with Transformers.

### Empirical Limitations

- Experiments are quite limited: It is good to have experiments where one can precisely simulate the theoretical conditions. So that is fine. But it is important to still carry out experiments in scenarios which are a bit more closer to the practical setup. Right now the only other setting is ViT on MNIST, and thereby makes for a very weak empirical evaluation.

- Could the authors carry out an empirical study of R during the course of training an proper Transformer on a language modelling or translation task? This would allow us to see when this particular issue with softmax starts appearing. Also, it would be good to conduct this with both Adam and SGD to draw effective comparisons.

References:

[1] Zhang, et. al. (2024). Adam-mini: Use fewer learning rates to gain more. arXiv preprint arXiv:2406.16793.

[2] Ormaniec, et. al. (2024). What Does It Mean to Be a Transformer? Insights from a Theoretical Hessian Analysis. arXiv preprint arXiv:2410.10986.

**Questions:**

Apart from the questions mentioned in the weaknesses section, I have the following question:

- Theorem 1 could be better presented. All the numerous terms are thrown at the reader, and without suitable explanation. Could the authors break it down and also better explain it? And also how each of the terms would scale so as to understand the various dependencies of the bound better.

- Why is there such a high variance (shaded region) in Figure 2?

- Also do you know why in Figure 2, without softmax, the R=1e25, ends up converging faster and to a lower training loss than R=10?

---

### Note · Authors · 2024-11-28

I have read and agree with the venue's withdrawal policy on behalf of myself and my co-authors.